# Executive Functions and Subjective Cognitive Decline: The Moderating Role of Depressive Symptoms

**DOI:** 10.3390/diagnostics15243164

**Published:** 2025-12-11

**Authors:** Marina Maffoni, Annalisa Magnani, Antonia Pierobon, Alessandra Mafferra, Fabrizio Pasotti, Carlo Dallocchio, Valeria Torlaschi, Daniela Mancini, Cira Fundarò

**Affiliations:** 1Psychology Unit, Montescano Institute, Istituti Clinici Scientifici Maugeri IRCCS, 27040 Montescano, Italy; antonia.pierobon@icsmaugeri.it (A.P.); valeria.torlaschi@icsmaugeri.it (V.T.); daniela.mancini@icsmaugeri.it (D.M.); 2Department of Psychology, Università Cattolica del Sacro Cuore, 20123 Milan, Italy; annalisa.magnani@unicatt.it; 3Neurology Unit, Department of Medical Area, ASST Pavia, 27058 Voghera, Italy; alessandra_mafferra@asst-pavia.it (A.M.); fabrizio_pasotti@asst-pavia.it (F.P.); carlo_dallocchio@asst-pavia.it (C.D.); 4Department of Rehabilitation, Civil Hospital of Voghera, ASST Pavia, 27058 Voghera, Italy; 5Neurophysiopathology Unit, Montescano Institute, Istituti Clinici Scientifici Maugeri IRCCS, 27040 Montescano, Italy; cira.fundaro@icsmaugeri.it

**Keywords:** subjective cognitive decline, executive functions, depressive symptoms, preventive and rehabilitation interventions, moderation analysis

## Abstract

**Background**: Executive dysfunction may be an early marker of neurodegenerative disorders, even if it is difficult to detect objectively. Subjective cognitive decline (SCD) may reflect subtle deficits in executive functions (EFs); however, SCD is also strongly influenced by affective factors such as depression. Whether depressive symptoms alter the link between EFs and SCD remains unclear. The present study tested whether depressive symptoms moderate the association between executive functioning and SCD. **Methods**: In this cross-sectional study, 65 outpatients completed a comprehensive assessment including executive functions (FAB), self-reported cognitive difficulties (CFI self-report), and depressive symptoms (PHQ-9). Descriptive statistics were performed, as well as a moderation analysis using PHQ-9 as moderator of the relationship between FAB and CFI self-report. **Results**: Participants (age: 69.28 ± 9.03) showed preserved EFs (FAB = 15.42, SD = 2.11), mild depressive symptoms (PHQ-9 = 6.57, SD = 5.02), and modest subjective difficulties (CFI = 4.87, SD = 2.57). The FAB main effect was positive but non-significant (β = 0.155, *p* = 0.266), while PHQ-9 was a significant positive predictor (β = 0.470, *p* ≤ 0.001). The interaction effect was significant (95% CI: [−0.166, −0.015], β = −0.343, *p* = 0.020). Specifically, simple slope analysis showed that at low levels of depression (−1 SD), better executive functioning was associated with higher SCD (β = 0.498, *p* = 0.039). Instead, the association was negative but non-significant at moderate (β = 0.155, *p* = 0.266) and high levels of depression (+1 SD) (β = −0.188, *p* = 0.229). **Conclusions**: In SCD, depressive symptoms are a stronger correlate of subjective cognitive difficulties than executive functions. Moreover, higher depression may modulate the executive functions–complaint link, reducing and potentially reversing it as symptom burden increases. Screening and treatment of depressive symptoms should be integrated into SCD assessment and care. Longitudinal and multicenter studies are needed to more deeply understand these preliminary results.

## 1. Introduction

Subjective Cognitive Decline (SCD) is defined as the self-reported experience of diminishing cognitive abilities without objective deficits identified by standardized neuropsychological tests [1,2,3,4].

This condition—which occurs in about a quarter of individuals over the age of 60 [5]—has become a major concern as it is increasingly considered an early manifestation of the Alzheimer’s disease (AD) continuum, which can lead first to Mild Cognitive Impairment (MCI) and then to severe dementia [2,6,7]. More precisely, SCD can precede MCI by up to 15 years and occur 10 years before a diagnosis of dementia [2,8]. Therefore, early identification of individuals at risk is necessary for implementing timely interventions that could prevent or delay the progression of cognitive decline [3,9,10].

The scientific literature shows inconsistencies about the relationship between SCD and objectively measured cognitive impairment, despite its potential significance as a precursor to neurodegeneration [3]. Although SCD has been associated with an increased risk of progression [2], most patients will not develop objective cognitive decline [6]. Indeed, the etiology of SCD is heterogeneous, and subjective complaints can be strongly influenced by a variety of psychosocial factors [11,12]. Thus, the clinical and research challenge is to differentiate SCD that represents preclinical neurodegenerative pathology from that associated with reversible medical causes or psychosocial disorders [3,13].

One of the most difficult challenges in the differential diagnosis of SCD is the frequent co-occurrence of affective symptoms, whose severity is often associated with a worse cognitive performance [3,12,14,15,16]. Focusing on depressive symptomatology, the association may reflect both a subjective amplification of cognitive difficulties [17] and age-related neurobiological effects—such as decreased gray matter volume and decreased white matter integrity [18,19,20]—resulting in increased susceptibility to neurodegenerative consequences [21]. Additionally, there is a nosographic overlap: concentration or thinking difficulties are diagnostic criteria for major depressive disorder [20,22], but they might be mistaken for the cognitive symptoms that characterize SCD, increasing the likelihood of a misdiagnosis. Regardless of the etiopathogenetic basis, longitudinal data suggests that SCD and depression independently predict the development of MCI and dementia, with the highest risk occurring when they coexist [23,24,25].

This interaction between affective and cognitive factors is not limited exclusively to memory, which has traditionally been the focus of SCD research [2]. Indeed, it has emerged that SCD may similarly involve other cognitive domains, including executive functions (EFs) [3,26,27]. It is well known that EFs are a class of higher-order cognitive processes that include a range of functions—such as attention, working memory, planning, inhibition, decision-making, and goal-directed behavior [3,27,28,29]. Within this framework, it is noteworthy that EFs, metacognition, and depression may be closely interconnected [30]. The infrastructure for metacognitive regulation and monitoring is provided by EFs. Specifically, processes such as inhibition, adaptability, and cognitive shifting might help individuals remain focused on tasks and appropriately calibrate confidence in their performance [26]. When these components are inefficient, attention could become biased toward negative self-referential content, error correction might be less effective, and accuracy judgments may tend to be overly pessimistic. These mechanisms could contribute to perseverative rumination, indecision, and self-assessment bias, with a wider gap between subjective complaints and objective performance [6,30,31]. This interpretation appears consistent with the literature on depression, in which executive deficits are associated with prefrontal circuit dysfunction and reduced fronto-subcortical white matter integrity, especially in older adults [32,33]. In this perspective, affective symptoms can amplify cognitive impairment through both metacognitive and neurobiological pathways [6]. However, the relationship between EFs, SCD and depressive symptoms remains poorly understood by the literature, and current evidence is far from conclusive.

Thus, the present investigation is based on these considerations as well as on the findings of another study we conducted and discussed elsewhere [34]. Briefly, in our previous paper, according to SCD criteria, the sample under investigation maintained objective cognitive performance across all areas, including executive functions [34]. Nevertheless, the sample showed mild depressive symptoms (PHQ-9), which were predictive of higher subjective decline (CFI). In detail, PHQ-9 was able to explain 9–11% of the variation in the self-perceived cognitive decline [34]. Since in that previous study only depression proved to be significant, we used this single variable in the present pilot moderation study, considering the small sample size.

Therefore, the aims and hypotheses outlined below are based on our previous observation of a significant interaction between cognitive profile and affective symptoms in individuals with SCD [34]. Specifically, our previous findings and literature evidence suggest that psychological variables—especially depressive symptoms—may significantly affect cognitive complaints in SCD, either causing new cognitive issues or making pre-existing ones worse. The coexistence of performance within normal limits and persistent complaints raises two questions that guide this study: (1) Are subclinical differences in executive functions linked to worse self-monitoring and more cognitive complaints, even when test findings are normal? (2) Does depressive symptomatology moderate this relationship, making the link between executive functions and subjective cognitive decline more pronounced at higher symptom levels? To answer these questions, a preliminary analysis is proposed in this study.

## 2. Materials and Methods

### 2.1. Study Design

This study is embedded in the ongoing MASCoD project (Multidimensional Assessment of Subjective Cognitive Decline). The screening tool has been described elsewhere [35], and full protocol details are available on ClinicalTrials.gov (Identifier: NCT05815329) [36]. In brief, MASCoD is a prospective, controlled clinical study comprising (i) a baseline assessment—neurological examination, brain ^18^F-FDG PET when clinically indicated, and comprehensive psychological and neuropsychological testing—and (ii) a 2-month technology-assisted cognitive training program.

More specifically—based on the preliminary results of the MASCoD validation [34]—the present investigation focuses only on the assessment phase in an attempt to delineate the joint profile of executive functioning and depressive symptomatology accompanying SCD and examines whether depressive symptoms may moderate the association between executive functioning and subjective cognitive decline.

The authors used Perplexity Pro (Perplexity AI, Inc., San Francisco, CA, USA), an AI-powered assistant combining large language models with real-time web search and source citations, exclusively for English language refinement. The conception, design, data acquisition, analysis, interpretation, and all manuscript content were independently developed by the authors, who also thoroughly reviewed and approved the final version.

### 2.2. Procedure

Participants were identified during routine neurological consultations within the Italian network of Centers for Cognitive Disorders and Dementias (CCDDs) [37]. Recruiting sites follow standardized Diagnostic Therapeutic Care Pathways (DTCPs) that align diagnostics, instrumental testing, rehabilitation, and patient care with international guidelines for cognitive disorders [38]. Patients with a clinical profile suggestive of SCD and without a previous diagnosis of cognitive impairment were identified by the attending neurologist. Consented participants underwent a multidimensional evaluation consisting of MASCoD screening, along with extensive neuropsychological testing and neuroimaging according to the standard usual care path.

Enrollment proceeded only after written informed consent was obtained. Additionally, participants were informed of their right to withdraw at any time without penalty, and no financial compensation was offered. This research adhered to the Declaration of Helsinki [39] and was approved by the ICS Maugeri Ethics Committee (Protocol CE 2666, 26 July 2022).

### 2.3. Participants

Participants (n = 65) were consecutively recruited from July 2022 at CDCDs—specifically at the Neurophysiopathology Unit of the IRCCS ICS Maugeri Institute (Montescano, Italy) and at the Neurology Unit of the Civil General Hospital of Voghera (Pavia, Italy)—where they were referred for neurological evaluation for suspected SCD.

The inclusion criteria considered for this study were the following: (i) self-reported SCD with no other diagnosed cognitive or neurological disorder, (ii) education within the Italian school system, (iii) adults aged 55 years or older, (iv) understanding of the study aims, and (v) voluntary, uncompensated participation. Instead, the exclusion criteria comprised: (i) serious medical conditions (e.g., severe cardiac or respiratory disease, active neoplasia); (ii) a current or past psychiatric diagnosis according to DSM-5-TR [22], (iii) prior diagnosis of cognitive impairment or dementia, (iv) clinically relevant visual/perceptual or hearing deficits, (v) illiteracy, and (vi) refusal or inability to participate.

Participants were not specifically debriefed concerning this study; rather, they were followed over time according to usual care and the standard DTCPs of the hospital, which ensured continuity of clinical management regardless of their research participation.

### 2.4. Measurements

All participants completed a comprehensive neuropsychological battery consistent with DTCP and Italian CCDD standards [38]. For the present analyses, we focused on the three below-described instruments.

Executive functioning was measured with the Frontal Assessment Battery (FAB) [40,41], a 6-item test assessing:

conceptualization/similarities—the subject is required to identify abstract links between objects from the same semantic category.phonemic verbal fluency—the subject generates as many words as possible within one minute that begin with a specified letter (e.g., “S”).motor programming—the subject is required to perform Luria’s “fist–edge–palm” sequence correctly six consecutive times.conflicting instructions—the subject gives the opposite response to the examiner’s alternating signals (e.g., If I tap twice, you tap once. If I tap once, you tap twice).go/no-go task—the subject again produces opposite responses but must now differentiate between two distinct response types (e.g., I tap once, you tap once. If I tap twice, you do not tap).environmental autonomy/prehension—the subject demonstrates the ability to inhibit automatic grasping when the examiner touches both of their palms.The FAB shows good psychometric characteristics, with an internal consistency of Cronbach’s α = 0.78, indicating a satisfactory degree of cohesion among the six subtests [41]. Each item is scored 0–3, with the raw total ranging from 0–18. Scores were adjusted for age and education to obtain equivalent scores. Lower adjusted scores indicate greater frontal/executive dysfunction. According to Italian norms, intact performance corresponds to an adjusted score ≥12.02.

Depressive symptoms were measured with the Patient Health Questionnaire-9 (PHQ-9) [42,43], a 9-item self-report questionnaire referring to what the person experienced during the past two weeks. Items are scored 0–3 (‘never’ to ‘almost every day’), with totals ranging from 0–27. Standard severity gangs were considered (0–4 ‘none’, 5–9 ‘mild’, 10–14 ‘moderate’, 15–19 ‘moderately severe’, ≥20 ‘severe’), with higher scores indicating greater depressive symptom burden.

Subjective cognitive decline was assessed with the Cognitive Functional Index (CFI) self-report [44]. The questionnaire includes 14 items, each of which has three different response modes: ‘yes’ (1 point), ‘maybe’ (0.5 point), ‘no’ (0 point). The total ranges 0–14, with a mean of 3.30 ± 2.44. Therefore, higher scores reflect greater self-perceived cognitive difficulties.

### 2.5. Statistical Analysis

All analyses were conducted using Jamovi software, version 2.6 [45]. Descriptive statistics (means and standard deviations for continuous variables; frequencies and percentages for categorical variables) were first computed to characterize the sample.

To test whether depressive symptomatology moderated the relationship between executive functioning and self-perceived cognitive decline, a moderated multiple regression model was estimated (see Figure 1). In this model, scores on the FAB were entered as the predictor (*X_i_*), PHQ-9 scores as the moderator (*Z_i_*), and CFI self-report scores as the outcome (*Y_i_*). The interaction term (*FAB × PHQ-9*) was included to assess moderation effects, as expressed by the following equation:

*CFI self-report_i_* = *b*_0_ + *b*_1_
*FAB_i_* + *b*_2_
*PHQ-9_i_* + *b*_3_
*(FAB_i_* × *PHQ-9_i_)* + *ε_i_*
(1)


Age was not included as a separate covariate since the FAB scores were already adjusted for age and education based on normative data [40], effectively controlling for these potential confounders at the measurement level.

Regression analyses were conducted using ordinary least squares estimation. Standardized coefficients (β), standard errors, t-values, and corresponding 95% confidence intervals were reported. To probe the interaction, simple slope analysis was performed to probe the conditional effect of FAB on CFI at the mean and ±1 SD of PHQ-9. Additionally, a Johnson–Neyman technique was applied. Statistical significance was set at *p* < 0.05 (two-tailed).

**Figure 1 diagnostics-15-03164-f001:**
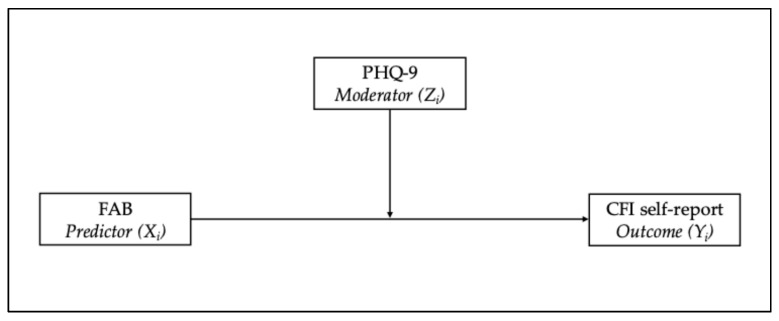
Moderated multiple regression conceptual model examining the moderating role of depressive symptoms (PHQ-9) in the relationship between executive functioning (FAB) and subjective cognitive functioning (CFI self-report).

## 3. Results

### 3.1. Socio-Demographic Profile

The sample comprised 65 outpatients, who were predominantly female (70.8%), retired (64.6%), and married or cohabiting (64.6%), with a mean age of 69.3 ± 9.03 years. Most participants had a lower-to-middle level of education (87.7% high school diploma or lower). Lifestyle and clinical characteristics were generally favorable, with low rates of current smoking (13.8%) and moderate alcohol use (12.3%). Physical activity was reported by 36.9%. Detailed sociodemographic, lifestyle, and clinical characteristics of the sample are presented in Table 1.

### 3.2. Neuropsychological and Psychological Performance

Higher scores indicate better performance on the FAB and greater symptom burden on the PHQ-9 and CFI. Results are reported in Table 2 and discussed below.

On the FAB, the sample showed preserved executive functioning (M = 15.42, SD = 2.11), well above the Italian threshold for statistically normal performance (≥12.02). On the PHQ-9, the mean score indicated mild depressive symptoms on average (M = 6.57, SD = 5.02), consistent with the conventional banding where values ≥5 reflect at least mild symptomatology. For the CFI self-report (n = 53, 12 missing), participants reported modest subjective cognitive difficulties (M = 4.87, SD = 2.57). This average lies below the study’s preliminary heuristic cutoff (≈5.74; mean + 1 SD), although variability suggests heterogeneity within the sample.

### 3.3. Moderation Analysis

The main effect of EFs on self-perceived cognitive functioning was positive but not statistically significant (b_1_ = 0.198, β = 0.155, *p* = 0.266). In contrast, the main effect of depressive symptoms was both positive and statistically significant (b_2_ = 0.250, β = 0.470, *p* ≤ 0.001), indicating that elevated depressive symptoms are associated with greater self-reported cognitive difficulties. Thus, depression serves as a stronger direct predictor of subjective cognitive difficulties compared to EFs. The interaction effect was significant too, suggesting notable interaction between EFs and depressive symptoms in predicting subjective cognitive difficulties (b_3_ = −0.090, 95% CI: [−0.166, −0.015], β = −0.343, *p* = 0.020). Particularly, the negative standardized regression coefficient indicates that the impact of EFs on cognitive functioning diminishes and changes in direction as depressive symptoms increase. Results are visually illustrated in Figure 2 and presented in Table 3.

To further explore the moderating role of depressive symptomatology in the relationship between EFs and self-perceived cognitive decline, this study categorized depressive symptomatology into low and high levels, defined as −1 SD below the mean and +1 SD above the mean, respectively. As presented in Table 4, the simple slopes analysis only found a significant association among participants with low depressive symptoms (–1 SD). In this group, better executive functioning was linked to higher self-reported cognitive difficulties (b = 0.635, 95% CI: [0.033, 1.237], β = 0.498, *p* = 0.039). Instead, the association was weaker and non-significant (b = 0.198, 95% CI: [−0.155, 0.551], β = 0.155, *p* = 0.266) at average depression levels. Conversely, among participants with severe depressive symptoms (+1 SD), the association tends to reverse—better executive functioning was related to fewer perceived cognitive problems—but this effect was not statistically significant (b =−0.239, 95% CI: [−0.634, 0.155], β = −0.188, *p* = 0.229). Overall, depressive symptomatology appears to modulate the link between executive functioning and subjective cognitive decline, diminishing and ultimately reversing its direction as depression increases.

To complement these conventional simple slopes, a Johnson–Neyman analysis was performed to identify the precise regions of significance of the PHQ-9 effect across the full range of EFs. As illustrated in Figure 3, the slope of depressive symptoms on CFI was significantly positive at lower levels of executive functioning. This suggests that lower EFs amplify the impact of depressive symptoms on subjective cognitive concerns. Beyond the Johnson–Neyman threshold (FAB = 1.18), the influence of depressive symptoms became non-significant as FAB increased.

## 4. Discussion

The present study sought to clarify the interplay between EFs, depressive symptoms, and self-perceived cognitive difficulties in aging populations, building on recent literature that highlights the complexities in metacognitive and subjective cognitive decline assessment [6].

In this light, our participants share features commonly reported in large international SCD cohorts—notably a female predominance and strong family as well as social support—including those in the Brain Health Registry SCD sample [47]. However, unlike some of those larger cohorts, our sample had comparatively lower education and different lifestyle habits—such as lower reported alcohol and tobacco use, strong emphasis on family networks [47]. Taken together, these patterns may likely reflect cultural, socioeconomic, and health-system influences that could shape the experience of SCD across the Italian population. Instead, from a clinical–neuropsychological standpoint, participants showed objective performance consistently above normative thresholds: mean FAB scores exceeded Italian cut-offs [40], and other domains were similarly intact as previously reported [34]. As expected in SCD, preserved test performance coexisted with persistent subjective concerns, with elevated CFI scores indicating self-perceived cognitive difficulties [44]. Additionally, mild depressive symptomatology emerged as a salient correlate. This trend is consistent with clinical and population-based data suggesting the relationship between subjective complaints and objective cognition is commonly influenced by depressive symptoms, and that direct connections usually attenuate or even disappear once mood is controlled [3,12,24].

Given the intact objective profile yet persistent subjective concerns—as well as the presence of mild depressive symptoms—we decided to understand whether perceived difficulties are better explained by executive control or by mood and whether mood moderates this link. In this regard, our findings showed that although executive functions had a positive main influence on self-perceived cognitive functioning, this effect was not statistically significant. In contrast, depressive symptoms showed a positive direct link, appearing to be a more robust predictor of subjective cognitive complaints than EFs in this sample. Importantly, the interaction between EFs and depressive symptoms was significant, suggesting that as depressive symptoms intensify, the impact of executive functioning on self-perceived cognition diminishes and may even change direction. As previously hypothesized by Cappa et al. [6], this pattern suggests that people who are able to retain both executive control and mood stability may have enhanced awareness or critical self-evaluation abilities. A nuanced hypothesis deriving from clinical experience is that if depressive symptoms are present, difficulties may be perceived more sharply or negatively, especially when cognitive abilities are still intact and capable of deciphering subtle changes. In this scenario, depression may serve as a persistent background tone that enhances self-critical perception. Conversely, when anxiety is present—a condition that often has acute effects, especially when optimal performance is needed, including cognitive performance, executive functions, even if globally preserved, may be perceived or experienced differently, potentially increasing feelings of inadequacy. However, anxiety was not tested here because, due to the small sample size, we only looked at the variable that was significant in our earlier work [34], which was depression symptoms. Nevertheless, current research suggests that metacognitive and insight-related studies should critically examine mood-related factors [48].

Despite the foregoing discussion, metacognition itself remains a domain with unresolved methodological challenges, as highlighted by Lehmann et al. [49]. Using trial-by-trial ratings of confidence across perceptual, memory, and attentional paradigms, their results suggest the existence of both domain-general and domain-specific components of metacognition [49]. This pattern calls into question the validity of self-report questionnaires as proxies for metacognitive efficiency, given their susceptibility to introspective bias and recall distortion [49].

These measurement issues also concern how self-perceived cognitive difficulties are assessed. For instance, in the context of the research on SCD, Gabbatore et al. [50] reported a significant negative correlation between subjective decline and pragmatic-linguistic abilities, underscoring the necessity for future research to provide more sensitive or preclinical cut-offs. Additionally, their findings pointed to declines in naming ability, working memory, selective attention, and cognitive flexibility, whereas basic linguistic comprehension and global cognitive functioning were largely unaffected [50].

Researchers are actively addressing the concerns outlined above. Novel cognitive testing measures are gaining prominence for their improved psychometric properties over established tests, especially for early detection of AD. The No Practice Effect (NPE) battery and Miami Computerized Functional Skills system showed better reliability and closer association with AD-related risk markers, further indicating the limitations of standard assessment tools in detecting subtle cognitive changes [51]. Additionally, semantic memory skills have been recognized as potential early markers for Alzheimer’s disease progression [52].

Notwithstanding the preceding discussion, a central controversy remains. Particularly, the correspondence between metacognitive measures and real cognitive performance is inconsistent, with evidence suggesting that subjective complaints are more closely connected to mental health status than objective cognitive decline. For instance, significant relationships have been documented between self-reported cognitive failures, depressive symptoms, optimism, and psychological well-being. Older adults with elevated subjective cognitive complaints consistently presented with less optimism, reduced well-being, and higher depressive symptomatology [53].

Taken together, our findings and recent evidence highlight the complex and multi-dimensional nature of subjective cognitive decline, the need to integrate mental health screening into cognitive impairment assessments, and the importance of developing tailored, sensitive tools that can distinguish between genuine cognitive changes and those driven by mood or metacognitive bias.

## 5. Strengths, Limits and Future Directions

The present study was designed, in its overall structure, with the aim of providing useful contributions to a deeper understanding of this topic. Firstly, it investigates the complex interaction between EFs, depressive symptoms, and subjective perception of cognitive decline, offering a significant contribution to the understanding of the multidimensional mechanisms involved in SCD. The emphasis on the moderating function of depression is another novel aspect. Indeed, the literature has shown that affective distress frequently serves as the primary cause of subjective cognitive complaints, possibly having a bigger effect than cognitive complaints like executive functions [3,12,14,15,16,17]. From this perspective, the suggested model advances a more thorough evaluation of the person and the variables influencing how they perceive their own cognitive functioning, consequently supporting the debate on the importance of integrating psychological and metacognitive screening tools into clinical practice and research on SCD.

However, the study has some limitations that must be considered. The results’ generalizability is limited by the small sample size, some missing values on the dependent variable, and the monocentric nature. A specific concern when conducting a moderation analysis with a small sample size is the reduced statistical power: small samples can lead to less reliable and unstable estimates of the interaction effects, making the results more susceptible to bias and less generalizable. Consequently, findings from moderation analyses with small samples should be considered preliminary and interpreted with caution, primarily serving to stimulate further research and critical reflection in the literature. Thus, these findings must be verified in bigger groups with a wider range of risk variables and sociodemographic backgrounds, as results obtained in a monocentric setting may not adequately represent groups with greater comorbidity or different levels of cognitive reserve. In addition, our investigation focused primarily on the FAB total score rather than analyzing individual subtest performances. While this approach aligns with the instrument’s original validation as a unitary screening measure for dysexecutive syndrome [40,41], it limits our ability to parse specific executive mechanisms (e.g., inhibition vs. shifting). However, we opted for the composite score to maximize psychometric robustness, as individual FAB subtests are coarse ordinal measures (0–3) that frequently exhibit ceiling effects and restricted variance in non-demented populations [54], which would likely compromise the reliability of complex moderation models.

Another methodological limitation includes the reliance on self-report measures for both affective and cognitive complaints, which may introduce shared method variance. Consistent with previous literature, we observed stronger correlations among self-reported scales than between subjective and objective cognitive measures [55,56]. However, by triangulating these subjective reports (PHQ-9, CFI) with standardized objective assessment (FAB), we aimed to mitigate this bias and provide a comprehensive view of the dissociation between perceived and actual cognitive function. This data may provide a more ecological window into the patient’s experience, reflecting not just their cognitive capacity (what they can do) but also their cognitive complaints (what they feel), which are clinically distinct but equally relevant constructs in this population.

Moreover, the absence of longitudinal assessments does not allow us to establish the causal directionality between depression and subjective complaints, an aspect that the literature indicates as crucial for understanding whether affective distress precedes or follows the perception of decline. A further limitation concerns the lack of specific measures for other possible variables that may interplay in self-awareness, such as pragmatic linguistic abilities [50] or personality traits. In fact, it is fairly well recognized that dimensions such as neuroticism and conscientiousness independently affect both the frequency and intensity of cognitive complaints and play a role in modulating metacognitive processes. Specifically, a systematic meta-analysis of 7642–10,564 participants found that higher neuroticism (r = 0.39) and lower conscientiousness (r = −0.36) were robustly correlated to increased cognitive complaints [57]. A 20-year longitudinal study further confirmed these associations, showing that higher neuroticism and lower conscientiousness predicted more cognitive complaints [57].

Looking ahead, multicenter and longitudinal designs, encompassing larger samples and people with diverse risk profiles, should be incorporated into future studies. This approach will allow for a more precise understanding of the SCD evolution, the role of depression, and the possible early and accurate identification of individuals at risk of progression to MCI or dementia. Moreover, it will be useful to verify the validity of the proposed model in contexts with numerous risk factors—such as family history, comorbidity, lifestyle, and cognitive reserve—to increase its external validity and predictive power. Likewise, future investigations should explore the role of more granular measures of executive functions, personality traits and metacognitive processes in the genesis and maintenance of cognitive complaints. The literature suggests that personality dimensions influence not only the propensity to complain but also subjective awareness of one’s cognitive state and the presence of any anosognosia. In this direction, the development of integrated assessment tools that combine subjective and performance-based measures represents a promising prospect. These tools would allow for a more objective assessment of metacognitive abilities and distinguish between forms of SCD linked to premorbid vulnerabilities and those associated with emotional biases, thus improving the capacity for early diagnosis and targeted prevention or rehabilitative intervention [58].

## Figures and Tables

**Figure 2 diagnostics-15-03164-f002:**
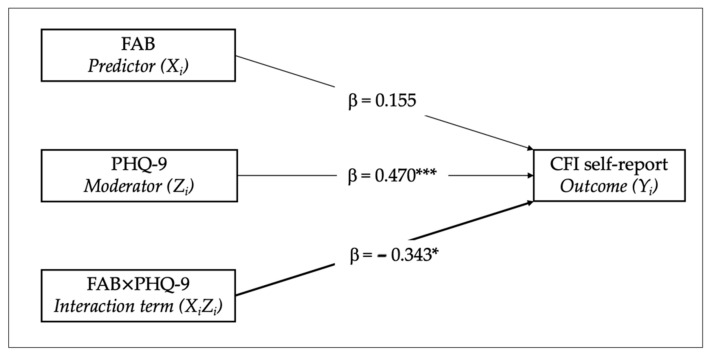
Moderated multiple regression statistical model examining the moderating role of depressive symptoms (PHQ-9) in the relationship between executive functioning (FAB) and subjective cognitive functioning (CFI self-report). Note: * *p* < 0.05; *** *p* < 0.001.

**Figure 3 diagnostics-15-03164-f003:**
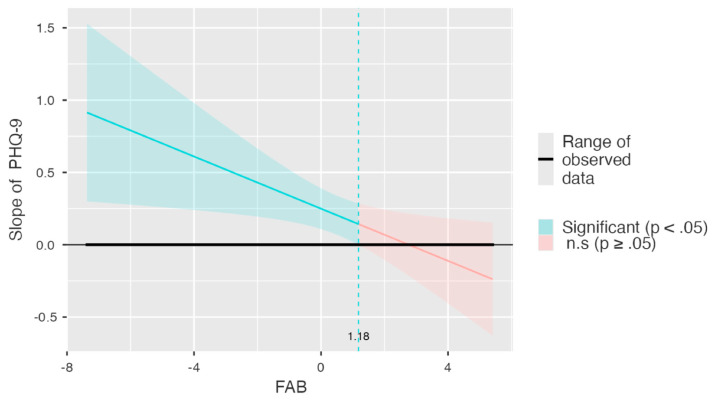
Johnson–Neyman plot illustrating how the effect (slope) of depressive symptoms (PHQ-9) on subjective cognitive functioning (CFI self-report) varies across the continuum of executive functioning (FAB).

**Table 1 diagnostics-15-03164-t001:** Descriptive characteristics of the sample.

Sociodemographic Variables			
**Variable**		**M ± SD**	**Min-Max**
Age		69.28 ± 9.03	55–84
BMI		25.79 ± 4.88	18.13–45.65
**Variable**	**Levels**	**n**	**% of Total**
Gender	Female	46	70.8%
	Male	19	29.2%
Study Title	Elementary school graduation	12	18.5%
	Junior high school	21	32.3%
	High school graduation	24	36.9%
	Bachelor’s degree	1	1.5%
	Master’s degree	5	7.7%
	Postgraduate specialization	2	3.1%
Marital status	Single	3	4.6%
	Married/Cohabiting	42	64.6%
	Widowed	12	18.5%
	Separated/Divorced	8	12.3%
Employment Status	Self-employed	4	6.2%
	Full-time employee	8	12.3%
	Part-time employee	4	6.2%
	Homemaker	4	6.2%
	Unemployed	3	4.6%
	Retired	42	64.6%
Socio-family support	Spouse/Partner	33	50.8%
	Son/Daughter	22	33.8%
	Parent	0	0%
	Other Family Member	2	3.1%
	Caretaker	0	0%
	Other No Family Member	0	0%
	Nobody	8	12.3%
**Variable**	**Levels**	**N**	**% of Total**
Physical Activity	Yes	24	36.9%
	No	23	35.4%
	No response	18	27.7%
Smoking	Yes	9	13.8%
	No	35	53.9%
	Former smoker	18	27.7%
	No response	3	4.6
Alcohol	Yes	8	12.3%
	No	52	80.0%
	No response	5	7.7%
Family History of Cognitive Disorders	Yes	30	46.15%
	No	30	46.15%
	No response	5	7.7%

Notes: M—mean; n—number of participants; SD—standard deviation; BMI = Body mass index.

**Table 2 diagnostics-15-03164-t002:** Neuropsychological and psychological data from our sample and normative value.

Variable	n	Missing	Normative Value	M ± SD
FAB	65	0	≥12.02	15.42 ± 2.11
PHQ-9	65	0	≥5.00	6.57 ± 5.02
CFI Self-report	53	12	≥5.74 ^1^	4.87 ± 2.57

Notes: M—mean; n—number of participants; SD—standard deviation. ^1^ A cutoff corresponding to +1 SD above the mean (3.30 + 2.44 = 5.74) was used to flag potential cognitive impairment [34,46].

**Table 3 diagnostics-15-03164-t003:** Moderation analyses.

			95% Confidence Interval					
	b	SE	Lower	Upper	β	t	df	*p*
FAB	0.198	0.176	−0.155	0.551	0.155	1.125	49	0.266
PHQ-9	0.250	0.070	0.108	0.391	0.470	3.552	49	<0.001
FAB × PHQ-9	−0.090	0.038	−0.166	−0.015	−0.343	−2.396	49	0.020

Notes: b—unstandardized regression coefficient; SE—standard error of the estimate; β—standardized regression coefficient; t—Student’s t-distribution; df—degrees of freedom; *p*—probability value (significance level).

**Table 4 diagnostics-15-03164-t004:** Simple slope estimates for the effect of executive functioning (FAB) on subjective cognitive functioning (CFI self-report) at different levels of depressive symptoms (PHQ-9).

			95% Confidence Interval					
	b	SE	Lower	Upper	β	t	df	*p*
Mean − 1 SD	0.635	0.300	0.033	1.237	0.498	2.119	49	0.039
Mean	0.198	0.176	−0.155	0.551	0.155	1.125	49	0.266
Mean + 1 SD	−0.239	0.196	−0.634	0.155	−0.188	−1.219	49	0.229

Notes: b—unstandardized regression coefficient; SE—standard error of the estimate; β—standardized regression coefficient; t—Student’s t-distribution; df—degrees of freedom; *p*—probability value (significance level).

## Data Availability

Raw data supporting the conclusions of this article will be made available by the authors, without undue reservation.

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
