# Peer review of "Executive Functions and Subjective Cognitive Decline: The Moderating Role of Depressive Symptoms"

_diagnostics, 2025, doi:10.3390/diagnostics15243164_

Round 1
Reviewer 1 Report
Comments and Suggestions for Authors
This was an interesting paper (albeit with a small N) exploring the links between executive function (EF), depression, and subjective cognitive concerns.
My main concerns relate to the small sample size of only 65 and why, given the age range, age was not included in the statistical modeling {although I see that FAB scores are age-adjusted}).
Other than those concerns, I have only a few minor points to raise. These points are as follows:
The start of the abstract needs to bring EFs out more to link to SCD and neurodegenerative disorders. Indicating the age of the sample would also help the reader. Please insert "to" between "needed" and "understand" or rephrase to "needed to more deeply understand these preliminary results".
Line 129: An explanation of what PerplexityPro is is required.
Line 147: Were participants debriefed afterwards?
Lines 166-172: A little more description of the individual FAB measures would be helpful to the reader.
Line 256: The N seems to be reported rather late and, at 65, is rather low.
Line 207: BMI does not seem to be defined (nor does it appear in the abbreviations).
Comments on the Quality of English LanguageThe manuscript is generally well written but there are a few points to address:
Line 142: Add "the" between "to" and "standard".
Line 218: Is "denied" the best choice of word in this context? Presumably this question relates to whether the participant reports themselves as being physically active (which could come out more here).
Line 288: Please correct "slop plot" to "slope plot".
Line 404-405: "quite known" does not sound quite right. "quite well known" or "fairly well recognised" or a similar phrase would be better.
Reviewer 2 Report
Comments and Suggestions for Authors
The authors present an interesting study showing that the self-reported cognitive decline is predicted by depression and the interaction of depression with executive functions (EF), but not by EF alone. The study is generally sound and the manuscript well-written. Hence, I believe that readers of diagnostics will appreciate to learn more about these findings.
Generally, it does not come much as a surprise to see that self-reports are more highly correlated than are objective performance measures with self-reports. Same-method bias should be discussed. However, the results are generally plausible.
Why did the authors only test moderation analyses for the FAB total scale? Was the computation of total scale warranted – please report the internal consistency in this sample? Typically, objective tests of EF are only very moderately related. Given the heterogeneity of tasks comprised in the FAB, why did the authors not test relations and moderation effects for specific tasks to infer which mechanisms drive the effect?
The sample is only N=53 not N=65, as there were 12 missing values in the dependent variable. This should be clearly communicated. However, the boot-strapping analyses demonstrate robustness of the findinsg, and future research will need to provide more evidence, as already discussed by the authors.
Apart from this, I have only few minor points and suggestions:
- Abstract: Please report standardized effect sizes (beta not b), also for non-significant effects.
- l. 166: Please provide internal consistency of the FAB score.
- p. 5 / bottom paragraph (socio demographics); l. 224: "All these data are likewise reported in Table 1": If this is the case, it would be sufficient to mention the Table in the main text instead of reporting all numbers again.
- Table 1: I am not sure if cumulative proportions are particularly informative. However, this column can stay if the authors prefer.
- l. 252 f.: "the negative unstandardized regression coefficient indicates that…": The standardized one would indicate the same, and the latter is easier to interpret. Generally, standardized effects should be reported (also betas in Tables 3 and 4).
- Simple slops analyses; l. 266: I know that simple slopes analyses are frequently reported to illustrate the interaction effect. However, I would recommend to visualize effects in the form of a Johnson-Neyman (JN) plot. This is more informative than the arbitrary (although conventional) regressions at moderator levels M +/- 1 SD.
- l. 377: "that may offer" [without s]; maybe this first sentence should be rephrased anyway as it sounds a bit stiff.
